# Thermal Tolerance of *Cyclops bohater* (Crustacea: Copepoda); Selection of Optimal and Avoided Conditions in Experimental Conditions



**Valentina Lazareva** [1,*], **Tatyana Mayor** [2], **Olga Malysheva** [1], **Elena Medyantseva** [1], **Svetlana Zhdanova** [1], **Andrey Grishanin** [1,3] **and Vladimir Verbitsky** [1]

[1] Papanin Institute for Biology of Inland Waters, Russian Academy of Sciences, 152742 Borok, Russia
[2] Limnological Institute Siberian Branch, Russian Academy of Sciences, Ulan-Batorskaya 3, 664033 Irkutsk, Russia
[3] Department of Biophysics, Faculty of Natural and Engineering Science, Dubna State University, Universitetskaya 19, 141980 Dubna, Russia
[*] Correspondence: laz@ibiw.ru; Tel.: +7-906-525-6036

**Abstract:** Temperature adaptations of ectothermic species as well as the plasticity of their thermal strategies are important for survival during temperature fluctuations, in particular, caused by global warming. The critical thermal maximum (CTM)—the values of the water temperature at which heat shock was noted (loss of motor activity in case of copepods) was determined under laboratory conditions. The "chronic" method was used to identify the temperature preferences of the copepods in which a group of test organisms are placed into a thermogradient apparatus. The main result is that in the experiment for individuals of the summer generation *C. bohater*, the optimal thermal conditions (FTP) were within 6–11 °C. Summer generation of this copepod in natural water bodies develops at a temperature of 5–12 °C, which is close to FTP in the experiment. At the same time, the thermal resistance of *C. bohater* (CTM 31.5 °C) was found to be the lowest among the species of the genus *Cyclops*. The differences between the thermal preferences of the winter and summer generations expand the temperature of normal performance (TNP) range and indicate a high physiological plasticity of the *C. bohater* population. This property is likely to allow *C. bohater* to survive as the climate continues to warm.

**Keywords:** thermal gradient; optimal temperature; temperature tolerance; copepod; phenology

## 1. Introduction

Environmental temperature directly affects all biochemical and physiological reactions in the organism of poikilothermic hydrobionts, which respond to thermal changes in the environment by avoiding harmful temperatures and choosing optimal temperature ranges [1,2]. Globally, water temperature determines the boundaries of ectotherms' ranges [3]. The seasonal sequence of species change in many cases can also be explained by changes in water temperature [4,5].

Recently, interest in the thermal adaptations of ectothermal species has expanded, since many data on global climatic changes have been accumulated. This is especially important for species inhabiting the Holarctic regions, where the greatest temperature fluctuations have been recorded [6,7]. Higher water temperatures may favor the development of thermophilic plankton species and/or species well adapted to stratified conditions [8]. In addition, according to the hypothesis by Cavieres et al. [9], under the conditions of global warming, it is possible to change the thermal strategies of species for survival in a wide temperature range.

Animals are physiologically most resilient in the range of preferred temperatures when their organism is exposed to minimal thermal stress [10,11]. This thermal preferendum (or

thermoregulatory behavior) is one of the main temperature reactions that characterize the adaptive capabilities of aquatic organisms. The thermopreferendum is very important—as a rule, it determines the characteristics of the distribution of animals in biotopes and their movement [12,13]. As shown for aquatic organisms, the final thermal preferendum (FTP) is innate and species-specific (review by [1]). FTP values usually coincide with the optimal temperature that animals need to move, grow and reproduce, choose habitats and optimize metabolic activity [14–17]. Little is known about the thermal resistance of freshwater copepods.

Critical thermal maximum (CTM) is a criterion of thermal resistance for poikilothermic vertebrates and invertebrates. This term is used to refer to both the method and the parameter itself. As a method, CTM is used in ecology to determine the temperature at which the first signs of physiological stress are noted [18,19].

The copepod *Cyclops bohater* (Copepoda: Cyclopoida) is a widespread poikliothermic hydrobiont. This freshwater species is found in mountainous lakes of southwestern Europe (Italy, France, Spain), from Croatia in the south to Sweden in the north [20–23]. In lakes, *C. bohater* prefers the hypolimnion area [20–27], but it can also be found in the near-shore areas [22–28]. In Russia, *C. bohater* is a rare and scarce species that lives in plain water bodies of the European part of the country between 55° and 60° N; here, it lives on the eastern margin (up to 38°34′ E) of its range. [24]. According to a review by Monchenko [21], all findings of *C. bohater* in Russia up to the early 1970s are questionable and require verification. A recent review by Holynska and Dimante-Deimantovica [22] also noted the lack of reliable data on the records of this species in Russia. This is a representative of cold-loving copepods; *C. bohater* reproduces at a water temperature of 7–11 °C in summer and less than 4 °C in winter [24–26].

The purpose of this work is field observations of thermal conditions of development of *C. bohater* in natural water bodies, taxonomic verification of its morphotypes by molecular analysis and determination of the range of selected and avoided temperatures in *C. bohater* representatives based on the results of experimental testing in a temperature-gradient installation. This study is the first in determining the range of selected and avoided temperatures in *C. bohater*.

## 2. Materials and Methods

### 2.1. Water Bodies in Which Field Distributions Were Analyzed

The development of the *C. bohater* population in nature was observed in three water bodies of the European part of Russia, located from 59°57′09″ to 55°45′11″ N and from 38°34′07″ to 38°27′30″ E (Figure 1). The Rybinsk reservoir is located in the Volga cascade of reservoirs and encloses the drainage basin of the Upper Volga [29]. Lake Ferapontovskoe belongs to the system of the Sheksna River—the left northern tributary of Volga (Upper Volga basin) [24]. Lake Glubokoe belongs to the Oka River basin—the right western tributary of Volga (Middle Volga basin) [24]. The studied water bodies differ in area (0.6–4550 km$^2$), maximum depth (21–32 m), and their location in the Volga River system (Table 1) but all belong to the basin of this river.

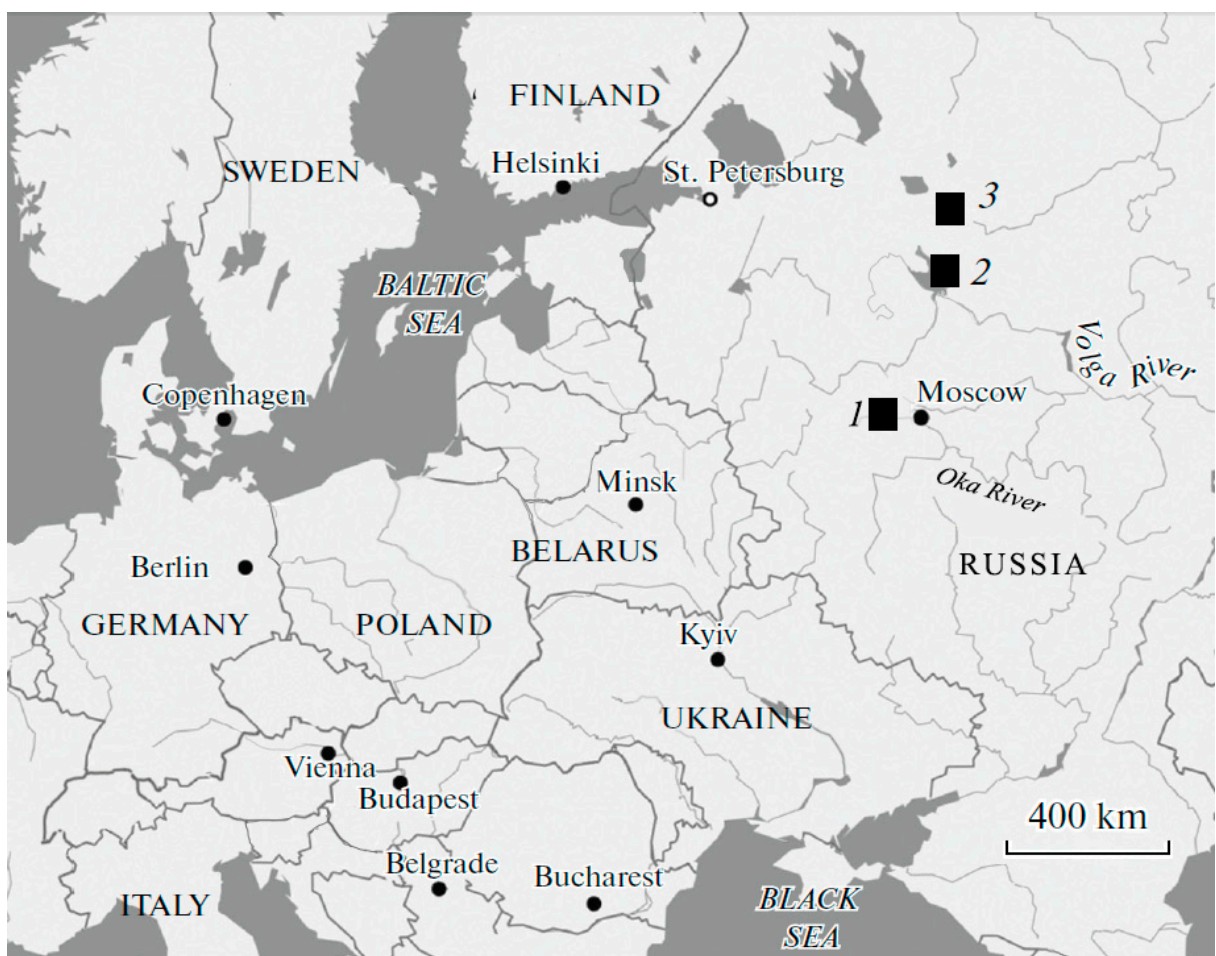

**Figure 1.** Records of Cyclops bohater in European part of Russia. 1—Lake Glubokoe, 2—Rybinsk Reservoir, and 3—Lake Ferapontovskoe.

**Table 1.** Description of water bodies, which were observed phenology of *Cyclops bohater*.

| Monogram | Toponym | Latitude (N) | Longitude (IE) | Area (km²) | Maximum Depth (cm) | Observation Times | Water Temperature (°C) |
|---|---|---|---|---|---|---|---|
| RR | Rybinsk Reservoir | 59°05′58″ | 38°27′30″ | 4550 | 2100 | December March | 0.3–1.7 0.7–3.4 |
| LF | Lake Ferapontovskoe | 59°57′09″ | 38°34′07″ | 1.5 | 3200 | February July | 1.1–1.5 8–11 |
| LG | Lake Glubokoe | 55°45′11″ | 36°30′18″ | 0.6 | 2700 | June–July | 5–9 |

In the Rybinsk Reservoir, regular annual monitoring was carried out at 6–20 sampling points from May to October in 2008–2014 and from December to March in 2009–2016. Lake Glubokoe was surveyed in July 2008 and the end of June 2021, and Lake Ferapontovskoe—in July 2013 and 2020. Observational data on Lake Ferapontovskoe in July 2007 and February 2009 are given according to Rivier [26].

### 2.2. Collection of Zooplankton in the Field

Copepods were collected using a small Juday net with a mouth diameter of 12 cm and a mesh size of 74 μm. The vertical distribution was studied in the deepest part of lakes (27 m depth in Glubokoe and 23 m in Ferapontovskoe). Vertical zooplankton tows were performed using a modified (closing) Juday net at four water layers in Glubokoe Lakes: 0–4 m (epilimnion), 4–9 m (metalimnion), 9–27 m (hypolimnion), and at the 2-meter layer above the bottom (near-bottom layer). Vertical tows of zooplankton were carried out in

three water layers in Lake Ferapontovskoe: 0–4 m (epilimnion), 4–9 m (metalimnion), and 6–23 m (lower water boundary of metallimnion + hypolimnion). We have information on the species reproduction from the two presented lakes only. In the Rybinsk Reservoir, copepods were collected from the bottom to the surface of the water using a plankton net. The *C. bohater* is found here only in winter (Table 1).

Samples were fixed with 4% formaldehyde. Species determination and linear measurements of animals were performed in the laboratory using StereoDiscovery V12 Carl Zeiss (Germany) and MS-2 LOMO (Russia) microscopes. Identification was performed in accordance with Monchenko [21], Holynska and Dimante-Deimantovica [22], Holynska and Dahms [30], Holynska [31].

### 2.3. Molecular Analysis

Additionally, the taxonomic affiliation of the copepods was checked by the method of molecular genetic analysis. For this, we used four specimens of *C. bohater* copepods captured in Lake Ferapontovskoe and fixed in 96% ethanol. Isolation of total DNA was carried out from somatic tissue in a double Encyclo buffer (without $Mg_2^+$) for PCR (Evrogen, Russia) containing 0.1 mg/mL of proteinase K. The mixture was incubated at 65 °C for 1 h, then heated at 94 °C for 10 min to inactivate proteinase K. The resulting hydrolyzate was stored at −20 °C. Fragments of mitochondrial DNA (mtDNA)—genes of the first subunit of cytochrome c oxidase (COI) and 12S rRNA (12S)—were selected as molecular markers. The fragments were amplified using the universal primers LCO1490 5'-GGTCAACAAATCATAAAGATATTGG-3' and HCO2198 5'-TAAACTTCAGGGTGACCA AAAAATCA-3', H13845-12S 5'-GTGCCAGCAGCTGCGTTA-3', L13337-12S 5'-YCTACTWT GYTACGACTTATCTC-3' [32,33].

The polymerase chain reaction of 20 μL mtDNA fragments was performed in Encyclo PCR buffer in the presence of 5 μM of each primer, 0.2 M dNTP, 0.5 activity units of the Encyclo Taq-DNA polymerase mixture, and 2 μL of a hydrolyzate containing DNA diluted in a ratio of 1:10. The temperature profile of the reaction included the following steps: 95 °C 4 min; 35 cycles 94 °C 15 s, 48 °C 15 s and 57 °C 15s for amplification of COI and 12S, respectively, 72 °C 1 min; 72 °C 4 min. Separation of amplicons was performed using gel electrophoresis in a single tris-acetate buffer without EDTA (0.04 M tris-acetate, pH 7.6). Pieces of gel with amplicons were sequentially excised, frozen, thawed, and centrifuged for 10 min at 12,000× *g*. The resulting eluate was used in the sequencing reaction. Determination of the nucleotide sequences of the target fragments was carried out in an eight capillary genetic analyzer ABI 3500 (Thermo Fisher Scientific, Waltham, MA, USA) using a BigDye Terminator Cycle Sequencing Kit v. 3.1 (Applied Biosystems, Austin, TX, USA). Alignment of nucleotide sequences and calculation of genetic distances were performed using the MegaX program [34]. The set of analyzed data includes nucleotide sequences borrowed from the GenBank databank [35,36]. The 12S nucleotide of *C. bohater* sequences of 395 and 456 bp (base pairs) in length were determined and placed in the GenBank database under numbers MW020535 and MW020536.

### 2.4. Zooplankton Collection for Testing

Copepods were sampled using a plankton net with a sieve mesh of 74 μm in Lake Ferapontovskoe (59°57′09″ N, 38°34′07″ E) on 19 July 2013 at a water temperature of 8–11 °C and on 5 July 2020 at a temperature of 8–10 °C. The samples were transported to the laboratory in a thermos bottle. The collected zooplankton was placed in Bogorov's chamber immediately after capture. Each individual cyclops was captured with a pipette under a microscope. At the indicated dates in Lake Ferapontovskoe, *C. bohater* was the only large dominant species of cyclops. The population was represented by females with eggs on 19 July 2013 and copepodids of 4–5 stages of development, as well as by adult females without egg sacs on 5 July 2020. After the experiment was completed, the copepods were fixed with 96% ethanol, measured and their species was additionally determined. The average body size, measured as the sum of the length of the cephalothorax, abdomen,

and caudal rami, was 2.20–2.40 mm for adult females of *C. bohater* and 1.60–1.76 mm for males. The body length of copepodids of the 4th stage of development (CopIV instar) was 1.22–1.48 mm and it was 1.56–1.76 mm for those of the 5th stage (CopV instar).

### 2.5. Description of Thermogradient Apparatus

Temperature preference was determined by the "chronic" method [1,17,37] in which a group of test organisms are placed into a thermogradient apparatus for 8 days. The design of the apparatus for measuring the FTP was a Herter tray with a metal floor and walls made of transparent Plexiglas with dimensions of 183 cm length × 10 cm width × 4 cm depth (Figure 2). The water from Lake Ferapontovskoe was filtered through a mesh plankton net of 75 μm and poured into the tray. The evaporated water was replaced each day by filtered lake water taken from the *C. bohater* habitat. The horizontal temperature gradient (from 3.8 ± 0.5 °C to 30.4 ± 0.7 °C) in tray was maintained using a thermostatic device TSS-1 (Russian Federation) with a heating element (0.8 kW) at one end of the tray and a cooling unit at the other end. To minimize convection currents and a vertical temperature gradient, the depth of the water in the tray was maintained at 10–12 mm depth. This design allowed for a horizontal temperature gradient of ~0.1 °C/cm. To measure and verify the water temperature, thermometers were placed permanently at fixed points along the tray every 10 cm. Measurements of the temperature at this point in the tray had a precision of 0.1 °C/cm. The number of copepods at a particular tray location within the thermal gradient was recorded using a linear scale with divisions of 1 mm. The water was uniformly illuminated by eight fluorescent lamps (40 watts, 120 cm, each). Two lamps were placed alongside the pan at a height of 0.45 m (Figure 2). Six additional ceiling lamps 1.4 m above the tray provided additional illumination. The illumination at the surface of the water was ~700 lux. Experiments were conducted under a photoperiod of 9: 15 h (light: dark) that was similar to the natural photoperiod in a lake at a depth of more than 10 m.

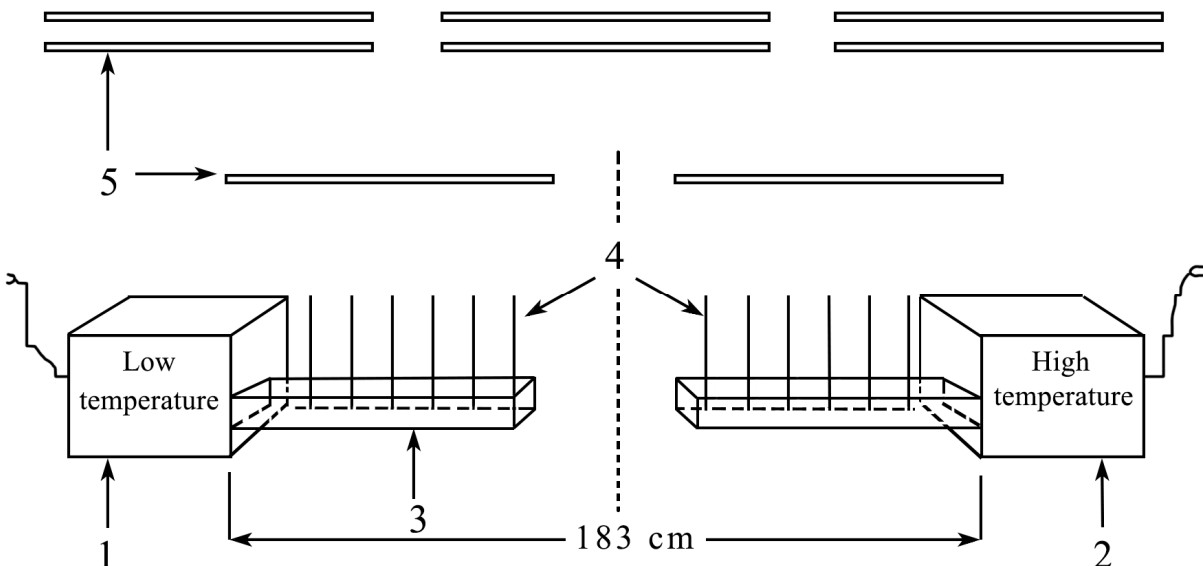

**Figure 2.** Schematic diagram of thermogradient apparatus. 1—refrigerating block, 2—heat block, 3—Herter tray, 4—thermometers, 5—eight lamps, by [38].

### 2.6. Experimental Procedures

The CTM method was first applied by Cowles and Bogert [39]; later, it was modified [40,41]. The CTM concept was finalized in the works [42,43]. The values of water temperature at which a heat shock was noted—critical thermal or thermal maximum (CTM) was considered the indicator of thermal resistance. For this, copepods (1 specimen) were placed in 15 mL test tubes with filtered water from the reservoir. A rack with test tubes

installed in a transparent cuvette with running water was connected to a thermostat. The initial temperature of the water in the cuvette (8° C) coincided with the temperature of the water from which the copepods were captured. The thermostat operated in the mode of increasing the temperature at a constant rate of 0.15 °C/min. The temperature increased by 25 °C within three hours. The final temperature was 33 °C, it was reached within three hours of the experiment. CTM was determined by the water temperature at the moment of interruption (loss) of locomotor activity of crustaceans [44,45].

On 20 July 2013, two series of CTM determinations were carried out. In each series, 9 specimens of *C. bohater* were tested, only sexually mature individuals were used; a total of 18 specimens were tested. According to the results of each series of determinations, the mean CTM value and standard deviation were found. After testing, the body length of all copepods was measured, and its arithmetic mean and error (SE) were calculated.

In 2020, the preferred (FTP) and avoidance temperatures and the temperature of normal performance (TNP) range were determined for *C. bohater*. The process of thermal selection is divided into two stages—transitional and stable selection [46]. In the initial period of stay in the temperature gradient, the distribution of animals depends on their state, determined by the previous thermal conditions. After a certain period of time, sufficient for reacclimation of animals to new conditions in the thermogradient, their distribution becomes stable. *Cyclops bohater* individuals were caught in Lake Ferapontovskoe on 5 July 2020, from 11:00 to 14:30 from a depth of 15–18 m, where the water temperature was 8–10 °C. From 23:00, the copepods were kept in a refrigerator at 10 °C. The study was carried out from 9 to 23 July 2020 (15 days) in the temperature range 1–31 °C, 40 *C. bohater* specimens were used in the experiment (mainly CopV instar). *Chlorella* sp. microalgae suspension was added daily at the end of the day at a concentration of $5.0–7.5 \times 10^5$ cells/mL, evenly distributed over the entire length of the chamber [38]. The readings were taken every 40 or 60 min, during the day from 8 to 18 h, 8–11 series of readings were carried out. A total of 146 series were carried out during which 4010 readings were recorded. Prior to each series of observations, thermometer readings were made to verify that any location did not vary more than 0.3 °C. Temperature was measured with digital thermometers placed equidistant. These experiments were carried out in the thermogradient apparatus. All other procedures were similar to those performed in the study of *Cyclops vicinus* [38].

*2.7. Calculation of Preferred and Avoided Temperatures*

In order to determine the preferred and avoidable temperatures, we used the methodology and criteria established by Diaz et al. [47] and Hernandez et al. [48]. To calculate the mean heat preference for each batch of samples, we used the arithmetic mean of the modal group of preferred temperatures [49]. To test the hypothesis that the data distribution is normal, we used Shapiro–Wilks W statistic [50]. We found the average of the preferred temperature for each day by averaging data from a series of samples. Then, we summarized the number of readings for all series for each degree, and based on these sums, we calculated the percentage distribution of copepods along the temperature scale. These data were used to calculate the ranges of modal values of selected temperatures, pessimal and avoidance temperatures. According to the W-criterion, the distribution of copepod individuals in the zone of selected temperatures during the entire observation time differed from the normal. Therefore, the preferred temperature was estimated not only by the arithmetic mean but also the median, lower (25%) and upper (75%) quartiles were calculated.

To assess the range of preferred temperatures, we used the values (modal range) at which >60% of *C. bohater* samples were found [38]. To establish the TNP range, we chose values at which >90% of *C. bohater* individuals were found. These conditions correspond to the natural temperature ranges in which individuals of native populations usually live, grow, feed, and reproduce [51]. Temperatures at which <10% of individuals were recorded were critical (or stressful). They determined the temperature at which organisms can survive but not thrive in nature. All calculations were performed using the R statistical package, version 3.2.2 [52].

### 3. Results

*3.1. Field Observation on the Distribution of C. bohater*

At present, *C. bohater* is found in only two small lakes of glacial origin (Lakes Glubokoe and Ferapontovskoe) and one large reservoir (Rybinsk Reservoir) (Table 2). In Lake Ferapontovskoe, the species reproduces in February at water temperatures below 1.5 °C and in July at 8–11 °C and in Lake Glubokoe, in June–July at 5–9 °C (Table 2). In the Rybinsk Reservoir, reproduction of C. bohater was recorded only in March under ice at a temperature of 0.7–3.4 °C. In summer, there is no thermal stratification of the water column in this reservoir; the water temperature near the bottom reaches 22 °C due to the shallow depth (average depth 5.6 m) and powerful circulation currents. A similar proportion (1:1 male/female ratio) of *C. bohater* in Lake Glubokoe at the end of June (Figure 3) indicated a recent start of breeding. On the contrary, in mid-February and July, in Lake Ferapontovskoe, the population was dominated by females (more than 55%), which indicated that the breeding season of *C. bohater* was about to end. Among juvenile specimens, copepodids of the 5th stage of development (CopV instar) prevailed in all cases. At the beginning of breeding (June 2021, Lake Glubokoe), these were probably maturing individuals of the winter generation. Whereas, at the end of breeding (February 2009 and July 2007, Lake Ferapontovskoe), these were CopV instars of a new generation, ready for the transition to diapause.

**Table 2.** The months and temperatures at which reproduction and rates of development of *Cyclops bohater* eggs in nature are maximal.

| Waterbody | Latitude (N), Longitude (IE) | The Period of Maximum Reproduction | T, °C for Eggs Development | Authors |
|---|---|---|---|---|
| Rybinsk Reservoir | 59°05′58″, 38°27′30″ | March | 0.7–3.4 | present study |
| Lake Ferapontovskoe | 59°57′09″, 38°34′07″ | July | 8–11 * | present study |
| | | July | 8–12 * | [26] |
| | | February–March | 1.1–5.0 | [26] |
| Lake Glubokoe | 55°45′11″, 36°30′18″ | June–July | 5–9 * | present study |
| Lake Wigry | 54°02′12″, 23°05′54″ | February | – | [20] |
| Lake Marien | 53°01′11″, 14°16′44″ | November–February | – | [25] |
| Lake Constance, | 47°36′06″, 09°27′09″ | January–February | – | [53] |
| Lake Zurich | 47°14′51″, 08°40′39″ | January–February | – | [53] |
| | | June–July | – | [20] |
| Lake Biel | 47°04′51″, 07°09′40″ | November–February | – | [54] |
| Lake Čingi-Lingi | 46°08′11″, 17°03′11″ | August–September | – | [55] |

Note. *—the water temperature in the hypolimnion of lakes is indicated.

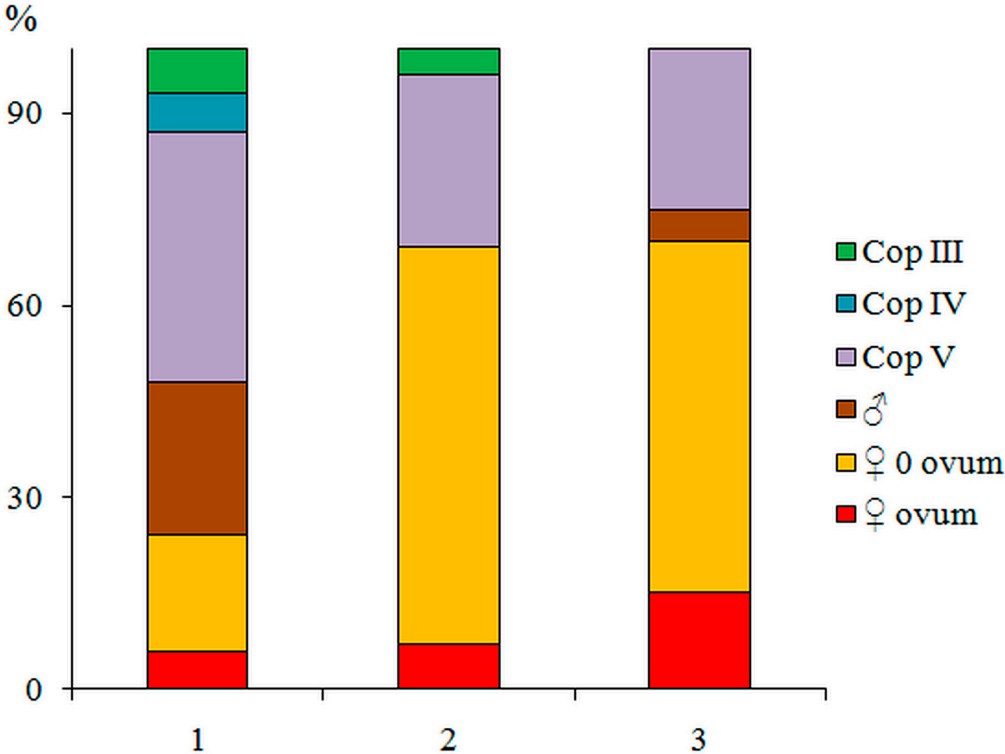

**Figure 3.** The structure of the breeding of *Cyclops bohater* population in lakes Glubokoe and Ferapontovskoe at different periods of the seasonal cycle. 1—Lake Glubokoe 28 June 2021, 2—Lake Ferapontovskoe 21 July 2007 and 3—in the same place February 2009 (2–3 by Rivier [26]). Cop III–Cop V—*C. bohater* copepodids (instar), ♂—male, ♀0 ovum—female without eggs, ♀ovum—female with eggs.

### 3.2. Molecular Data

For three out of four *C. bohater* individuals from Lake Ferapontovskoe, the COI fragment was not amplified; 12S amplification was more efficient. Analysis of the obtained nucleotide sequences showed that they are identical and genetically closest to the nucleotide sequences of the populations of *C. bohater* from Germany (KP773186, KP773187) and *C. strenuus lacustris* from Norway (KP773204), which are available in the GenBank database (Figure 4). The values of genetic distances between two populations of *C. bohater* from Germany and Russia, as well as between *C. bohater* and *C. strenuus lacustris* vary within 0.9–1.2%. The values of the genetic divergence of the 12S gene between two populations of *C. bohater* from Germany and Russia correspond to the intraspecific level known for this genus [23,35]. The genetic distances between the nucleotide sequences of *C. bohater* and *C. abyssorum* are 22.3–25.6% and correspond to the interspecies level. Thus, the molecular data are consistent with the identification of *C. bohater* by morphological characters. The studied specimens of the genus *Cyclops* from Lake Ferapontovskoe belong to the species *C. bohater*.

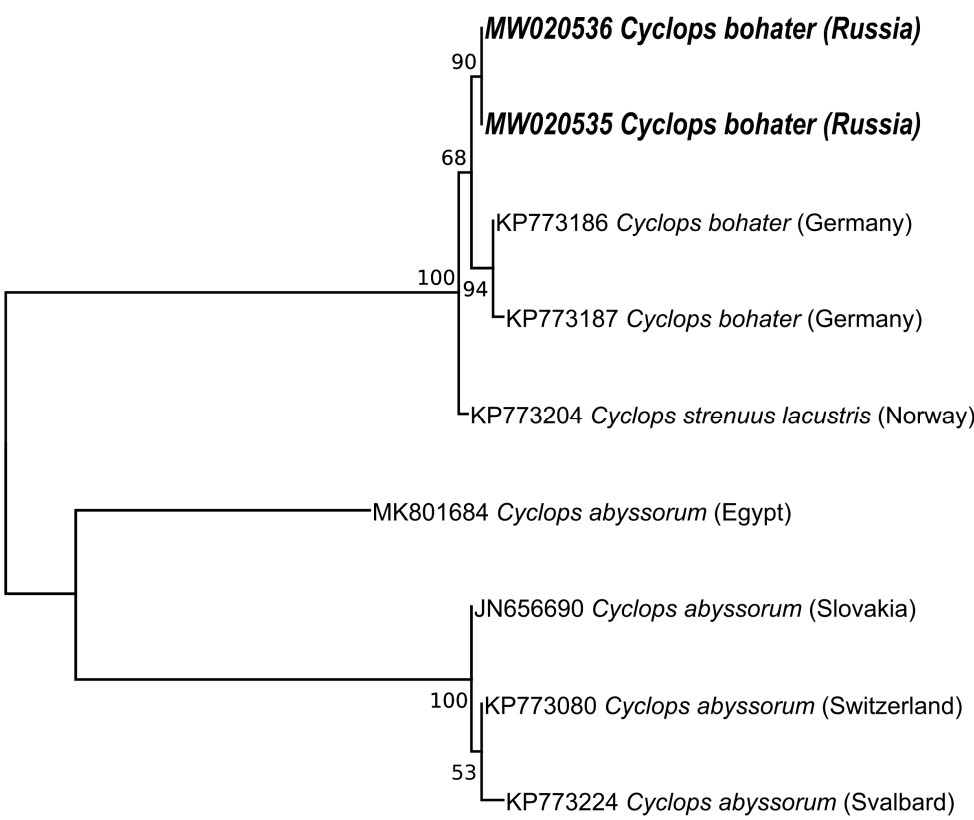

**Figure 4.** Phylogenetic tree constructed based on the 12S gene fragment by the maximum likelihood method (ML, T92). The number in the node is the bootstrap value of the branching node support. Nucleotide sequences obtained in this study are marked in bold.

*3.3. Experimental Data*

3.3.1. CTM Determination

Copepods *C. bohater* (18 specimens with a body length of 2.0 ± 0.05 mm) were captured on 19 July 2013 at a water temperature of 8–9 °C. The average CTM value was 31.5 ± 1.02 °C. The CTM varied from 29.6 °C to 32.9 °C for different specimens of *C. bohater*, with the difference reaching 3.3 °C.

3.3.2. Determination of FTP and TNP

For *C. bohater*, the transition period was 9 days; for comparison, for *C. vicinus* and *C. strenuus*, it was 7 days each [38–45]. During this time, individuals of *C. bohater* were distributed either in a wide temperature range from 7.9 ± 1.9 °C to 10.4 ± 2.4 °C (during 1–3 and 7–9 days), or (during 4–6 days) they found two zones of preferred temperatures 8.7 ± 2.4 °C (50–65% of readings) and 17.7 ± 1.2 °C (20–40%) (Table 3, Figure 5a,c).

**Table 3.** Mean, median, range, lower and upper quartiles of preferred temperatures *Cyclops bohater*.

| Time, Days | PT$_m$ (°C) | Median of PT (°C) | Lowe Quartile of PT | Upper Quartile of PT | Range of PT (°C) | Readings in Range of PT (%) | TNP (°C) | Readings in Range of TNP (%) | W-Criterion | *p*-Value |
|---|---|---|---|---|---|---|---|---|---|---|
| 1 | 10.2 ± 1.8 | 10 | 9 | 12 | 8–13 | 73 | 8–17 | 94 | 0.919 | <0.001 |
| 2 | 7.9 ± 1.9 | 7 | 6 | 9 | 6–12 | 70 | 6–20 | 94 | 0.833 | <0.001 |
| 3 | 9.9 ± 1.8 | 10 | 9 | 11 | 7–14 | 77 | 7–20 | 99 | 0.868 | <0.001 |
| 4 | 8.5 ± 1.6 | 8 | 7 | 10 | 7–12 | 64 | 7–19 | 100 | 0.822 | <0.001 |
|   | 18.6 ± 0.5 | 19 | 18 | 19 | 18–19 | 23 |   |   |   |   |
| 5 | 8.8 ± 1.7 | 8 | 7 | 10 | 7–12 | 54 | 7–19 | 99 | 0.890 | <0.001 |
|   | 17.9 ± 1.0 | 18 | 17 | 19 | 16–19 | 38 |   |   |   |   |
| 6 | 8.8 ± 2.4 | 8 | 7 | 11 | 6–13 | 59 | 6–19 | 98 | 0.939 | <0.001 |
|   | 16.6 ± 1.2 | 17 | 15 | 18 | 15–18 | 23 |   |   |   |   |
| 7 | 9.7 ± 2.7 | 9 | 7 | 11 | 6–15 | 82 | 5–18 | 96 | 0.940 | <0.001 |
| 8 | 9.9 ± 3.3 | 10 | 7 | 12 | 5–16 | 84 | 5–19 | 94 | 0.940 | <0.001 |
| 9 | 10.2 ± 2.9 | 10 | 8 | 13 | 6–15 | 78 | 6–19 | 95 | 0.954 | <0.001 |
| 10 | 8.0 ± 1.6 [a] | 9 | 7 | 9 | 5–10 | 63 | 5–17 | 92 | 0.916 | <0.001 |
| 11 | 7.3 ± 2.5 [a] | 8 | 5 | 9 | 3–11 | 72 | 3–17 | 99 | 0.964 | <0.001 |
| 12 | 8.9 ± 1.6 [a] | 9 | 7 | 10 | 6–11 | 63 | 6–17 | 92 | 0.959 | <0.001 |
| 13 | 8.4 ± 1.8 [a] | 9 | 7 | 10 | 6–11 | 68 | 6–19 | 93 | 0.874 | <0.001 |
| 14 | 8.0 ± 1.5 [a] | 8 | 7 | 10 | 6–10 | 66 | 5–17 | 92 | 0.798 | <0.001 |
| 15 | 7.3 ± 2.0 [a] | 8 | 6 | 9 | 4–10 | 77 | 4–16 | 94 | 0.894 | <0.001 |

Note: PT$_m$—mean preferred temperatures, TNP—the temperature of normal performance. All range of samples, W-criterion—Shapiro–Wilks W statistic, [a]—values PT$_m$ and median of PT equal FTP.

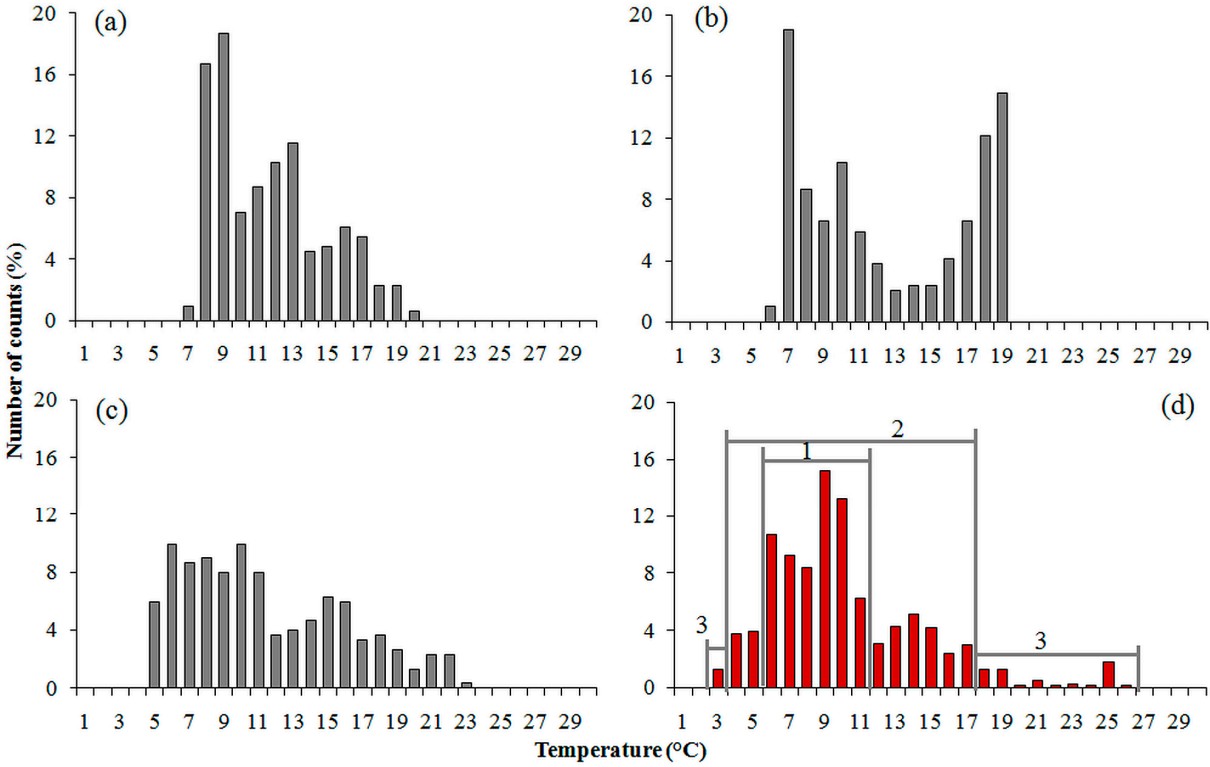

**Figure 5.** The distribution of the counts of individual *Cyclops bohater* at each location in the temperature gradient in 9–23 July 2020. (**a**)—Day 1, (**b**)—Day 5, (**c**)—Day 8, (**d**)—Mean during Day 10–15. 1—The Final Thermal Preferendum (FTP) equals the Optimum Thermal Conditions (OTC); 2—Temperature of normal performance (TNP); 3—Pessimal Temperatures; Red highlights the final result.

Starting from the 10th day of the experiment, 63–77% of readings corresponded to the temperature range of 3–11 °C. The final preferred temperature (FTP) was 8.5 ± 1.6 °C (mean ± standard deviation), median 9 °C (Table 3). During the period of stable selection (within 10–15 days), the average range of chosen temperatures was 6–11 °C (Figure 5d). It

accounted for 63% of the counts. The temperature range of 4–17 °C is defined as the zone of normal activity (TNP, 94% of readings on average). Copepods avoided temperatures of 0–2 °C and above 26 °C. These temperature values can be classified as pessimal (or stressful). Under these conditions, the summer generation of *C. bohater* from which the crustaceans were taken for the experiment can live but in a depressed state. However, in winter this species breeds under ice at a temperature of 1–3 °C (Table 2), which is pessimal for its summer generation.

## 4. Discussion

In nature, *C. bohater* breeds in cold water: in winter under ice and/or in summer in the cold hypolimnion of lakes [20,25,26,28]. In lakes Constance (Lake Constance), Zurich (Lake Zurich), Mindelse (Lake Mindel), and Mariense (Lake Marien), *C. bohater*, as a rule, has one breeding period in winter in January–February, and in May it goes into diapause on the fifth copepodite stage of development [22,25]. Only one winter breeding season of this species was also observed in the Rybinsk Reservoir [24]. In other water bodies, *C. bohater* may have a second breeding season, which is observed in the cold hypolimnion in summer [21,26,28,55]. In the studied Lake Ferapontovskoe, this species bred twice a year in February and July [24]. In copepods, males are the first to reach sexual maturity, but also the first to die off at the end of breeding [21,26]. It is known [25]) that *C. bohater* goes into diapause at the CopV instar stage.

Prior to the systematic analysis [22], *C. bohater* was mixed with morphologically similar *C. lacustris* and *C. abyssorum*. Thus, in Lake Ferapontovskoe, *C. bohater* was previously identified as *C. abyssorum* [26]. According to [35,56], *C. bohater* is genetically very close to *C. lacustris*. Our results of molecular analysis confirm this. In the phylogenetic reconstruction based on the analysis of morphological characters [23], these two species form one group ("divergens-clade") with the European *C. divergens*. *C. bohater* morphologically reliably differs from *C. lacustris*, as shown by a number of studies [22,24]. According to Holynska and Wyngaard [23], the "abyssorum" ("abyssorum-clade") variety group is sister to the "divergens" ("divergens-clade") group, which includes *C. bohater*. Our molecular analysis confirmed that Lake Ferapontovskoe from which specimens were taken for the experiment, is inhabited by *C. bohater*, and not by *C. abyssorum*. According to the results of experiments presented in this work, the copepod inhabiting the lake belongs to the species *C. bohater*.

In our experiments, we used specimens of the summer generation of *C. bohater* caught in July. The optimal thermal conditions (FTP) for them were in the range of 6–11 °C. In natural water bodies, the summer generation of *C. bohater* develops at 5–12 °C [24,26,28]. This is close to the FTP range obtained in our experiment. Our experiment showed that individuals of the summer generation of *C. bohater* avoid temperatures below 3 °C and above 26 °C. We did not find any literature data on the habitation of this species in water bodies at water temperatures above 20 °C. Unfortunately, most works [22,25] do not provide the values of water temperature under the ice at which this species breeds in winter. However, there is evidence that in some years the winter generation of *C. bohater* in Lake Ferapontovskoe bred at a temperature of 1.1–1.5 °C [26]. We assume that the FTP for the winter generation of *C. bohater* is lower than for the summer generation. Note that the shift of CTM and thermal optimum (FTP) depending on the habitat temperature is known for another species of this genus, *C. strenuus* [45]. A linear relationship between thermal resistance and environmental temperature is observed in many ectotherms [6,57,58].

The CTM values of *C. bohater* (31.5 °C) fall within the range of those of other Cyclopoida (30.7–33.1 °C) that breed in cold water under ice (0–5°C) or in summer in the hypolimnion of lakes at 5–12 °C [45,59]. The average CTM of *C. bohater* is close to that of other cold-loving Cyclopoida species. The thermal resistance of *C. bohater* is one of the lowest among Cyclopoida; its CTM is closest to that noted for the stenothermic species *Megacyclops gigas* (31.9 ± 0.99), which breeds in winter or early spring at 4–9 °C [59]. Thus, for *C. insignis*, *C. kolensis*, *C. strenuus*, and *Megacyclops gigas*, the CTM varies from 30.7 ± 0.61 to 33.1 ± 0.73 °C [59].

Behavioral and evolutionary mechanisms of thermal resistance are crucial for protecting ectotherms from extreme temperatures during a warming climate, while aquatic animals have greater plasticity in thermal resistance than the terrestrial [60]. An increase in CTM in *Daphnia* has been experimentally shown in modern times under conditions of higher water temperature [6]. Our experiment showed that the summer generation of *Cyclops bohater* chooses a higher temperature than that at which its winter generation breeds in the same waterbody. This confirms the ideas [60] on the high physiological plasticity of aquatic animals.

There is also a relationship between the thermal resistance of ectotherms and geographic latitude. According to Sunday et al. [7], extremely low temperatures strongly decrease at high latitudes (up to 60° N) for all animals (ecto- and endotherms), while the limits of heat tolerance in freshwater animals moderately decrease from south to north. Our data indicate that the TNP limits for *C. bohater*, a generally cold-water species (FTP 6–11 °C), are shifted towards higher temperatures (up to 17 °C). This supports the hypothesis [9] that species may exhibit plasticity of thermal strategies in new global warming scenarios. In particular, the results obtained suggest that *C. bohater*, at least its summer generations, has a high potential for shifting FTP (by 5–6 °C) to the high temperature region, which will allow this species to survive with further climate warming.

In general, despite the low thermal resistance of *C. bohater*, its thermal preferences in the experiment and in natural water bodies indicate a relatively wide TNP range (1–17 °C) and high physiological plasticity of this copepod. The summer generation of *C. bohater* has a particularly high potential for shifting FTP (by 5–6 °C) to the region of high temperatures. This property is likely to contribute to the survival of *C. bohater* as the climate further warms.

**Author Contributions:** V.L. and V.V. conceptualized and designed the experiment; O.M., E.M., A.G. and S.Z. conducted the experiments and collected the data; V.L., O.M., T.M. and S.Z. analyzed the data and made the figures; V.L. wrote the manuscript with suggestions from T.M., O.M., and S.Z. All authors have read and agreed to the published version of the manuscript.

**Funding:** This study was carried out within the Framework of The State Assignment No. 121051100109-1. The results of Section 3.3 were carried out within the Framework of the State Task No. 0279-2021-0005 and supported by Russian Foundation for Basic Research (RFBR) grant No. 19-07-00322a.

**Institutional Review Board Statement:** Not applicable.

**Informed Consent Statement:** Not applicable.

**Data Availability Statement:** The datasets used and/or analyzed during the current study are available from the corresponding authors on request. Sequences were deposited in GenBank under the accession numbers mentioned in the text.

**Acknowledgments:** We thank Maxim V. Zagoskin for their constructive comments.

**Conflicts of Interest:** The authors declare no conflict of interest.

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
