# Peer review of "Thermal Tolerance of Cyclops bohater (Crustacea: Copepoda); Selection of Optimal and Avoided Conditions in Experimental Conditions"

_diversity, doi:10.3390/d14121106_

Round 1
Reviewer 1 Report
The manuscript by Lazareva et al. describes the optimal and critical temperature ranges at which the poikilothermic crustacean Cyclops bohater can thrive and reproduce. The study combines field observation in three Russian lakes, temperature experiments, and a literature survey. The observed thermal plasticity of this species typically occurs at its optimal temperature of 5-12°C but has an upper resistance temperature of up to 31.5°C might allow surviving as water temperature continues to warm with respect to climate change.
The manuscript is nicely and clearly written. The field observations and experiments are appropriate and well-explained. Please find my comments below that are intended to improve the readability of the manuscript.
Major comments
1) I highly recommend keeping the same structure in objectives, methods, results, and discussion. The authors included in their manuscript: (1) field observation, (2) a literature survey, (3) taxonomic verification of morphotypes, and (4) experimental testing of temperature tolerance.
2) I gathered from the first part of the results section that the authors performed a literature survey on the occurrence and distribution of C. bohater in addition to their field observations and experiments. This literature survey needs an explanation of the methods and could already be highlighted in the objectives and introduction of the study. Please see also minor comments below.
3) Apart from the literature survey, the results section should focus on the description of your own results and citations should be reduced to a minimum. All references to background knowledge should be moved elsewhere. Please check the results section carefully. I mention many issues below.
Minor comments
Title: I suggest changing „pessimal“ to „critical“ or “avoided”.
L. 41-43 Sentence is hard to understand. Please rephrase.
L. 44-52 I suggest restructuring this paragraph and focusing more on organisms and less on definitions. The last sentence could be moved to the beginning of the paragraph: “Animals are physiologically most resilient in the range of preferred temperatures when their organism is exposed to minimal heat stress (). This thermal preferendum …”
L. 56 Please add “physiological stress”.
L. 56-58 These two sentences don’t tell much to the reader. I suggest either moving them to methods or giving them more information on how the method looks like and what exactly changed by the modification or finalization.
L. 59 I suggest rephrasing to “A widespread poikliothermic hydrobiont is the copepod Cyclops bohater (). This freshwater species is found in mountainous lakes of Southwestern Europe (Italy, France, Spain), from Croatia in the south to Sweden in the north (). However, …”
L. 65-66 This sentence on a general habitat of the species could follow the first sentence of the paragraph.
L. 67-70 The objectives of the study need more elaboration. Are there published experimental results on the temperature range of this species? If not, please state that your study is the first in determining the range of selected and avoided temperatures in C. bohater. If yes, please refer to this existing knowledge and how your study fills a gap.
L. 87-100 Please add information on the collection of zooplankton in Rybinsk Reservoir.
L. 106 Change “belonging” to “affiliation”.
L. 188-189 Sentence is a repetition and could be removed. The information from the sentence “The temperature increased by 25°C …” could be moved to the previous paragraph.
L. 193-204 Please add that these experiments were carried out in the thermogradient apparatus.
L. 217-218 Change “samples” to “individuals”
L. 221 I suggest changing “pessimal” to “critical” if that fits.
Results. I suggest the same order as in the methods. Results of taxonomic verification could be moved following the field observation data.
L. 225 Change header to “Field observation” or “Field observation on the distribution of C. bohater”
L. 226-227 This sentence does not describe your own results and it fits better with the introduction or discussion.
L. 227-229 Is there information on monitoring of other lakes in Russia that support the point of absence of C. bohater in other Russian lakes?
L. 230-246 and Figure 3. Information on the Rybinsk reservoir missing. Or do you have information on species reproduction from the two presented lakes only? If yes, please state in methods.
L. 251 “Experiment 1” is not necessary for the header.
L. 255-257 This information belongs to the discussion.
L. 265-267 Information belongs to methods.
L. 268-272 This information either belongs to methods or maybe introduction.
L. 275-277 Please add the unit to temperature values.
L. 277-281 Information belongs to methods.
L. 300-303 Information belongs to methods.
L. 336 This information would be good to mention in the introduction already.
L. 368 Suggest removing “strongly”
Author Response
Author's Notes to Reviewer 1
I thank the referee for a careful analysis of our article and important comments. All comments of the reviewer were taken into account.
Major comments
- Structure in objectives, methods, results, and discussion is brought to the same order: (1) field observation, (2) a literature survey, (3) taxonomic verification of morphotypes, and (4) experimental testing of temperature tolerance.
- 2. Literature review from results section moved to discussion, methods, or introduction.
- The results section has been carefully checked. All references to basic knowledge have been moved to another location. We took into account all the minor comments.
Minor comments
Title: Changed „pessimal“ to “avoided”.
- 41-43: Sentence is rephrasing.
- 44-52: Paragraph structure changed.
- 56: “physiological stress” added.
- 56-58: These two sentences have been moved to methods (section 2.6).
- 59: Sentence is rephrasing.
- 65-66: Sentence moved.
- 67-70: The objectives of the study are rephrasing. Added that “This study is the first in determining the range of selected and avoided temperatures in C. bohater”.
- 87-100: Information on the collection of zooplankton in Rybinsk Reservoir added.
- 106 Changed “belonging” to “affiliation”.
- 188-189 A repeat has been removed. The sentence “The temperature increased by 25°C …” moved to the previous paragraph.
- 193-204 Added that these experiments were carried out in the thermogradient apparatus.
- 217-218 Changed “samples” to “individuals”.
- 221 Changed “pessimal” to “critical”.
Results. The structure of the results section has been brought into order with that of the methods section.
- 225 Changed header to “Field observation on the distribution of C. bohater”
- 226-227 this sentence have been moved to the introduction.
- 227-229 Yes, there is information on monitoring of other lakes in Russia that support the point of absence of C. bohater in other Russian lakes. This information has been added to the introduction.
- 230-246 and Figure 3. We have information on the species reproduction from the two presented lakes only. It's included in the methods (section 2.2).
- 251 “Experiment 1” removed.
- 255-257 this information moved to the discussion.
- 265-267 Information moved to methods (section 2.6).
- 268-272 this information moved to methods (section 2.6).
- 275-277 the unit to temperature values added.
- 277-281 this Information moved to methods (section 2.7).
- 300-303 this information is a repeat, it has been removed.
- 336 This information moved to the introduction.
- 368 “strongly” removed.
December, 8, 2022 Valentina Lazareva
Reviewer 2 Report
The manuscript is interesting and provides valuable information on thermal tolerance of the ciclopoid Cyclops bohater (Crustacea: Copepoda) in experimental conditions. In the attached text (pdf) I indicate some suggestions and details to modify in the references.
I recommend its publication with minor revisions.

Author Response
Author's Notes to Reviewer 2
I thank the referee for a careful analysis of our article and important comments. All comments of the reviewer were taken into account, with the exception of L 236.
- Section 2.3: Molecular analysis is included in the objectives.
- L 155-156: The method of water preparation for the experiment has been clarified.
- L 171: Photoperiod of 9: 15 h (light: dark). Added that the schema similar to the natural photoperiod in a lake at a depth of more than 10 m.
- L 197-198: Real concentration 5.0–7.5 × 105 (10 E+5), bibliographical reference is made.
- L 236-237: “In copepods males are the first to reach sexual maturity, but also the first to die off at the end of breeding [26, 31].”
It would be more correct to say that their survival is lower than that of females.
But I can't say that, in the cited articles there is no data on the survival of males and females.
“A high proportion of C. bohater males (1:1 male/female ratio) in Lake Glubokoe at the end of June (Figure 3) indicated a recent start of breeding.”
Rather than a high proportion of males, it would be more correct to indicate that there is a similar proportion of males and females.
This sentence has been changed.
- References 53 and 61 changed.
December, 10, 2022 Valentina Lazareva